# Determinants of Pro-Environmental Behaviour in the Workplace

**Bob Foster** [1,*], **Zikri Muhammad** [2,*], **Mohd Yusoff Yusliza** [2], **Juhari Noor Faezah** [2], **Muhamad Deni Johansyah** [3], **Jing Yi Yong** [4], **Adnan ul-Haque** [5], **Jumadil Saputra** [2], **Thurasamy Ramayah** [6,7,8,9,10,11] and **Olawole Fawehinmi** [2]

1   Faculty of Economics and Business, Universitas Informatika dan Bisnis Indonesia, Bandung 40117, Indonesia
2   Faculty of Business, Economics and Social Development, Universiti Malaysia Terengganu,
    Kuala Nerus 21030, Terengganu, Malaysia; yusliza@umt.edu.my (M.Y.Y.); faezahjuhari95@gmail.com (J.N.F.);
    jumadil.saputra@umt.edu.my (J.S.); olawolefawehinmi@umt.edu.my (O.F.)
3   Faculty of Mathematic and Natural Sciences, Universitas Padjadjaran, Bandung 45363, Indonesia;
    muhamad.deni@unpad.ac.id
4   Faculty of Business and Law, School of Management and Marketing, Taylor's University, 1, Jalan Taylors,
    Subang Jaya 47500, Selangor, Malaysia; sereneyong@outlook.com
5   Faculty of Business, Yorkville University, 100 Woodside Ln, Fredericton, NB E3C 2R9, Canada; ahaque@yorkvilleu.ca
6   School of Management, Universiti Sains Malaysia, Gelugor 11800, Pulau Pinang, Malaysia; ramayah@usm.my
7   Department of Information Technology and Management, Daffodil International University, Dhaka 1341, Bangladesh
8   Faculty of Economics and Business, Universiti Malaysia Sarawak, Kota Samarahan 94300, Sarawak, Malaysia
9   Faculty of Business and Management, Universiti Teknologi Mara, Shah Alam 40450, Selangor, Malaysia
10  Faculty of Economics and Management, Universiti Kebangsaan Malaysia, Bangi 43600, Selangor, Malaysia
11  Faculty of Accounting and Management, Universiti Tunku Abdul Rahman, Petaling Jaya 46200, Selangor, Malaysia
*   Correspondence: bobriset@unibi.ac.id (B.F.); zikri@umt.edu.my (Z.M.)

**Abstract:** The primary notion of sustainable development is to maintain a promising future for the planet and the next generation by raising the awareness of sustainable development of people around the world. This study seeks to foster and enhance more sustainable behaviour in households, workplaces, schools, and higher educational institutions; previous research has placed increasing attention on the identification of factors of pro-environmental behaviour. Accordingly, this study aims to examine the elements influencing the pro-environmental behaviour of employees in the workplace. A survey was performed from January to February 2020 on 150 public employees of an organisation in Terengganu. Out of 150 employees, only 84 participated and had their responses collected by using convenience sampling. Smart PLS-SEM was used in analysing the relationships between the variables. The result of this study found that green lifestyles have a significant positive effect on pro-environmental behaviour. However, the impacts of environmental commitment, environmental consciousness, green self-efficacy, and green human resource management were insignificant. This study provides data that were developed using a cross-sectional design; the assessment of causality among the constructs was a risky process. Furthermore, the study collected data from a single source, namely the employees, which would enhance the relationships through common method bias. The findings of this study also offered several managerial implications for green organisations.

**Keywords:** environmental commitment; environmental consciousness; green lifestyles; green self-efficacy; green human resource management; pro-environmental behaviour

## 1. Introduction

The main idea of sustainable development is to maintain a promising future for the planet and the future generations. Environmental issues are perhaps the most relevant part among the 17 sustainable development goals (SDGs) proposed by the United Nations (UN) [1]. According to Singh, Pradhan, Panigrahy, and Jena [2], the most popular definition of "sustainable development" was put forward to the United Nations (1983) by the Brundt-land Commission as to "meet the needs of the present and future without compromising the ability of future generations." Based on the goals of the 2030 Agenda of sustainable

development, Goal 12 mentioned "Responsible Consumption and Production" to increase the awareness of sustainable development in terms of resource consumption among people around the world. It also aims to promote a healthy lifestyle [3]. Based on the researchers' investigation of the negative impact of human beings on the carrying capability of Earth in the last four decades, WWF [4] and Blok, Wesselink, Studynka, and Kemp [5] found that the increasing contamination of water, land, and air resources, including the decrease in natural resources, were caused by human behaviour [5,6]. Over the past decade, a global consensus has been reached with regard to the importance of reducing the impact of human activities on the environment [7]. Furthermore, Stern [8] and Chen, and Chen, Huang, Long, and Li [9] stated that following the proof that human activities are among the major causes for environmental problems, pro-environmental behaviour (PEB) could be considered important for the future.

Many countries have formed policies for pollution management in industries by reducing greenhouse gas emissions while preserving natural resources from depletion. Previous research focused on the development of a sustainable lifestyle in households, companies, and educational institutions [5]. However, not only is environmental protection related to national policies, but individuals' awareness of and behaviour with regard to environmental protection, that is, their PEB, also need to be enhanced [7]. In the world of business, the initiatives of the organisations to introduce internal green plans, execute environmental management systems, and apply certifiable standards have become inefficient when proper employee integration is not implemented [10,11]. Notably, active participation from employees could positively influence the success in the integration of environmental standards and policies with ISO 14001. However, some organizations are faced with challenges in promoting several types of behaviours, such as recycling, switching on and off electrical appliances, choosing video conferencing as a replacement for travelling, and using public transportation to promote cleaner production (i.e., prevent the production of waste) and reduce the organisation impacts on the environment. Although these activities might seem irrelevant on the individual level, they could possibly have a significant influence on the organisation environmental performance [12].

Protecting the environment through human activities is known as "PEB", "green behaviour", "environment-friendly behaviour", or "low-carbon behaviour" [13,14]. Meanwhile, Graves and Sarkis [15] defined PEB as a set of environmental responsibilities, such as improving knowledge related to the environment, creating green products and processes, and reviewing actions harmful to the environment. For instance, employees conserving energy by switching off unnecessary electrical appliances, using stairs instead of the lift, avoiding single side paper printing, minimising waste, and generating ideas for environmental protection [16]. Although numerous studies were performed to describe similar types of sustainable activities or PEB, the focus was placed solely on reducing the negative impact of human behaviour on the environment [5]. Moreover, PEB has been specified by Stern [8] and Fu, Zhang, and Bai [13] into two categories, namely private (e.g., buying, using, disposing of personal services and products) and public PEB (e.g., pro-environmental rules and policies, encouraging people to join pro-environmental activities, and addressing environmental issues). Additionally, psychologists and sociologists have been attempting to disclose the elements influencing PEB in the workplace settings for the past 30 years.

The participation of employees in the environmental initiative is a complex subject, as PEB is rather a voluntary action instead of compulsory. Compared to households, limited exploration has been performed on behaviours in the workplace, despite their presence [17]. It was reported in several empirical studies that the behaviour of employees regarding corporate greening was connected to pollution hindrance, efficient environmental management systems, and green innovations, although there is no confirmation regarding the exact nature of the employees' participation in corporate greening [17]. According to Yuriev, Boiral, Francoeur, and Paillé [10], there is no definite concept of PEBs for employees, while Scherbaum, Popovich, and Finlinson [18] and Fawehinmi, Yusliza, Wan Kasim, Mohamad, Sofian Abdul Halim [19] defined it as the eagerness to engage in pro-environmental ac-

tivities, such as double-sided printing, avoiding single-use cups, reducing waste, helping organisations to execute green strategies, and developing ideas to address environmental problems. Notably, besides the importance of employees' PEB in the promotion of environmental performances [20], their participation addresses environmental issues and is a good strategy for organisations with environmental responsibilities to improve environmental performance [19,21,22]. In this way, employees try to match their values with that of the organization's pro-environmental values, thus creating a sustainable organizational climate, where employees are likely to be engaged in PEB [23].

In collaboration with this concept, Li et al. [1] elaborated that human activity is one of the major driving factors of climate change, which has led to a worldwide consensus that PEB must be encouraged. According to Yuriev et al. [10], managers often neglect the behaviours not described in official documents. Daily, Bishop, and Govindarajulu [24] also highlighted that the role of employees to reduce the environmental footprint from the organisations is neglected by the higher-ups due to unpredictable behaviours. Therefore, PEB at the workplace depends on efficient human resource management, which is challenging to achieve only through formal approaches [10]. Based on several empirical studies on the differences between green practices at home and the workplace, it was found that the same person recycles more frequently at home than in the workplace [10,25]. Moreover, Lo, Peters, and Kok [26] showed significant differences between energy-saving behaviours inside and outside the office. Norton, Parker, Zacher, and Ashkanasy [27] stated that the PEB-related challenges were associated with the individuals' characteristics, while other behaviours were associated with the organisation. Recently, Faraz, Ahmed, Ying, and Mehmood [28] indicates that fostering PEB can reap strategic advantages to organizations by lowering costs, enhancing revenue, developing a positive image, attaining sustainability initiatives, and the maintaining a competitive edge.

Following previous findings, a question regarding the actual factors of sustainable behaviour and methods to enhance the behaviour has been raised. Furthermore, this article begins with an overview of the literature regarding the links between environmental commitment, environmental consciousness, green lifestyle, green self-efficacy, green human resource management (HRM) and PEB. This is followed by the hypotheses to be tested in this study. The research method is discussed, followed by a discussion of the analysis and results. Several original contributions were made in the literature and practice in two ways. First, a theoretical contribution was made in this study through the implementation of the Norm-Activation model to examine the elements influencing the PEB. Second, this study has also contributed to the extant literature by analysing the factors of PEB, among employees in a specific organization in Malaysia.

## 2. Literature Review

### 2.1. Norm-Activation Model and Pro-Environmental Behaviour

The norm-activation model (NAM) was developed to investigate and elaborate on the factors influencing PEB among employees. Onwezen, Antonides, and Bartels [29] referred to the NAM as altruistic behaviour and environmentally sustainable actions. Therefore, self-conscious emotions are relevant for the understanding of PEB within the NAM. This theory is also known to be effective in the identification of environmentally-responsible decision making and practices [30]. Notably, NAM has been commonly used in research on different forms of pro-environmental intent or behaviour, while PEB is perceived as a pro-social behaviour [31]. According to the NAM, people are more likely to reduce their energy consumption when they feel morally obliged to do so, in other words, when they experience a strong personal norm to save energy [32]. Once the managers identify reducing environmental impacts as the responsibility of enterprise, they are more willing to take active green practices to promote enterprises to practice green sustainability [33]. It has been argued that the higher the level of an individual's moral obligation, the stronger his or her intentions are to engage in pro-environmental behaviour [34], which would lead to high environmental commitment. Furthermore, if they have high degrees of self-confidence,

individuals can perform complicated engagements as challenges, rather than seeing them as threats that should be ignored [35].

According to Confente and Scarpi [36], the theory is based on awareness of consequences and ascription of responsibility. It posits that awareness of a problem is an antecedent of responsible behaviour [36]. Therefore, if an employee possesses values that are pro-environmental in nature, their level of environmental consciousness is greater [37]. Finally, Saeed, Afsar, Hafeez, Khan, Tahir, and Afridi [38], referring to the NAM, argue that when employees are aware of environmental problems, they are more likely to exhibit pro-environmental behaviours. Only when people believe that collective efforts can solve the climate change problem are their intentions to engage in active PEB [34]. In this theory, the main crux lies in the need to generate awareness of harmful effects on people with regard to the environment. Thus, it is believed that an individual's intention to engage in pro-environmental behaviour to mitigate climate change is related to an individual's levels of self-efficacy and collective efficacy beliefs when confronted with a global problem such as climate change [34]. The norms in the organization could be the presence of a well implemented green HRM practices, which gives employees an idea of the organisational value. Hence, employees would align their behaviours according to the green organizational values. This study was predicted to contribute to the emerging body of literature and develop a comprehensive theoretical framework for the involvement of the employees in PEB. It also has the potential of offering useful insights on improving the practices of environmental activities among employees in the organisation to address future employee challenges.

### 2.2. Environmental Commitment and Pro-Environmental Behaviour

Commitment could be expressed as a promise or assurance to behavioural actions [39]. According to Afsar and Umrani [40], environmental commitment is known as a state of mind, internal temperament, and psychological condition representing individuals' sense of duty and obligation to the environmental issues in the workplace. Meanwhile, environmental commitment leads to contentment towards the environment, investment in the behaviours of general ecological, and the willingness to perform actions for the benefits of the environment [41,42]. According to Rahman and Reynolds [42], people with high environmental commitment, which is also known as biospheric values, are prepared to take any actions for the safety of the environment. In turn, employees become inclined to pay attention to and endorse prioritized organizational issues [43]. It seems consistent to hold that an employee demonstrates commitment to the environment when he or she has the desire to share, identify with and care about the environmental concerns of his or her organization [44].

Employees with enthusiasm in the environmental concerns would participate in pro-environmental activities and influence other employees towards participation. Once encouraged, the employees would voluntarily perform PEB without being instructed by the managers or higher-ups. In supporting this notion, the study by Oreg and Katz-Gerro [45] applied a comprehensive example of participants from 27 different countries to identify the environmental commitment and determine the relationship between environmental matters and several PEBs, such as recycling, environmental citizenship, and preventing the use of cars. Meanwhile, several studies, such as a study by Mayerl and Best [46], examined the willingness to engage in PEB, including attitudes, behaviours, and intention of protecting the environment. Another investigation was conducted by Wan, Shen, and Yu [47] on the determinants of willingness to support recycling activities in Hong Kong. Within this literature, empirical findings have consistently shown the safety orientation displayed by safety-specific transformational leadership (i.e., another type of target-specific transformational leadership) can influence subordinates' safety climate perceptions such that employees believe safety is prioritized over other organizational issues [43].

It was proven by Ito, Leung, and Huang [48], that environmental commitment had a significant impact on PEB. Melo, Ge, Craig, Brewer, and Thronicker [49] indicated that beliefs and behaviours towards the (natural) environment led to individuals' interests in

pro-environmental activities. The degree of the individuals' beliefs regarding the impact of their habits on the environment also led to a similar outcome. Furthermore, voluntary organisational pro-environmental practices are based on the employees' knowledge of environmental concerns, environmental management programmes, and greening strategies of companies. This notion was in line with the finding by Afsar and Umrani [40], which highlighted that when employees are proud of their organisation when they perceive it as socially conscious. Subsequently, their desire to identify with the company contributes to a higher commitment to the organisation. Accordingly, the following hypothesis was developed.

**Hypothesis 1 (H1).** *Environmental commitment has a positive effect on PEB.*

*2.3. Environmental Consciousness and Pro-Environmental Behaviour*

Environmental consciousness refers to the interests and concerns of the environment, the attitudes applied to reduce environmental issues, and the fact that this consciousness is among the significant factors of human behaviour [50]. de Vicente Bittar, [51] mentioned previous literature, which theoretically highlighted the environmental-related factors of human behaviour. Notably, environmental consciousness offers more information about environmental factors, behaviour, and attitude [50].

Meanwhile, the norm-activation theory (NAT) proposes that when the environmental issues experienced by an individual is witnessed by others, they will consider their contribution to those issues and make conscious decisions regarding the environment. Employees may also feel responsible when the company is challenged by issues related to the environment, and subsequently suggest solutions. Further, in a very recent attempt, Thormann and Wicker [52] demonstrated the positive effect of environmental consciousness on PEB among active sport club members in Germany. The study by Zientara and Zamojska [53] also highlighted the positive effect of environmental consciousness on PEB and hypothesised that an individual's level of environmental consciousness was higher with the presence of pro-environmental values [37]. Subsequently, other views regarding the environment and the perceived value of environmental impacts would increase the perceptiveness of environmental knowledge [54]. Accordingly, the following hypothesis was generated:

**Hypothesis 2 (H2).** *Environmental consciousness has a positive effect on PEB.*

*2.4. Green Lifestyle and Pro-Environmental Behaviour*

Lifestyle has been described in the research for economics and transportation as a demographic variable (e.g., income, travel availability, and population). Therefore, it could be used interchangeably for subjects related to sustainable consumption [55,56]. Generally, the acceptance towards the applications for lifestyle in previous years was not positive, which indicated the way consumers and social groups distinguish between behaviours and motives. A person who is not exhibiting a PEB (or lifestyle) is less likely to receive such social stigma because it is seen to be socially understandable (excusable) if a person does not adopt such a lifestyle [57]. Furthermore, the adoption of green lifestyles for satisfactory, simple, and sustainable consuming is possible as a part of the green economy. However, extending this lifestyle would be costly, although this cost could be reduced when certain aspects of well-being are omitted [58]. To illustrate this situation, in the tight framework of the economic standard, higher income and consumption expenditure equate to a higher wellbeing, while reducing one's expenditures for a sustainable life will lead to inconveniences [59]; this framework leads to the challenges faced by employees or individuals in their voluntary involvement with PEB at the workplace or home.

In the United Kingdom, there is significant interest among policymakers in the search of the ideal tools to transition to greener lifestyles [60]. A green lifestyle is applying green behaviours into one's daily practices [61]. According to a report by Binder and Blakenber [58], a connection between life satisfaction and a green lifestyle was found because of

self-image and an environmentally friendly characteristic of an individual instead of PEBs, such as recycling and conserving water [62]. Meanwhile, Fraj, and Martinez [62] stated that lifestyle had a direct impact on environmental behaviours. With the establishment and changes in the crucial cognitive elements, behavioural changes could propagate in the environments through all aspects of an individual's lifestyle [63]. Accordingly, the following hypothesis was developed:

**Hypothesis 3 (H3).** *A green lifestyle has a positive effect on PEB.*

### 2.5. Green Self-Efficacy and Pro-Environmental Behaviour

Self-efficacy is defined as a personal evaluation for one's potential to develop motivation, rational resources, and the behaviour required to cope with the forthcoming situation [2,64]. In generally, self-efficacy belief is low if an individual believes themself to be capable only in distinct situations and for very few behaviours [2]. It is also perceived as a mechanism affecting pro-environmental spill over, which is an effect of the increase in the probability to commit other PEBs due to the existing commitment to one PEB (e.g., Lanzini and Thøgersen [65]; Lauren, Fielding, Smith, and Louis [64]) despite the limited understanding of the spill over among PEBs. Self-efficacy reflects confidence in one's ability to control their motivation, behaviour, and social environment [66]. Furthermore, environmental self-efficacy is related to the individuals' beliefs that they are capable to reduce the harmful effects [67]. Meinhold and Malkus [68] indicated that a higher level of self-efficacy in a person allowed the classification of the types of individuals with a higher probability for positive attitudes and behaviours toward the world.

In promoting PEBs, various studies aimed to determine environmental beliefs, value, self-efficacy, or effectiveness, which influenced the environmental behaviour of individuals [67,69]. In addition, Singh et al. [2] highlighted that individuals were assumed to have more or less firm self-beliefs in different tasks or particular domains and specific situations. It was found that self-efficacy led to several PEBs, such as recycling (e.g., Tabernero and Hernández [70]) and using eco-friendly shopping bags (e.g., Lam and Chen [71]). Subsequently, it was suggested that people with higher self-efficacy could be inspired by PEBs to invest more effort in keeping up with the behaviours (Lauren et al., 2016). Furthermore, Choong, Ng, Na, and Tan [72] demonstrated the positive effect of green self-efficacy on PEB. This finding was in line with Huang's study [67], which indicated that higher self-efficiency in specific tasks show higher capabilities and trust to perform the task, and it may increase a person's behavioural intention. Therefore, the following hypothesis was developed:

**Hypothesis 4 (H4).** *Green self-efficacy has a positive effect on PEB.*

### 2.6. Green Human Resource Management and Pro-Environmental Behaviour

Green HRM is the incorporation of environmental consciousness in the overall HRM recruitment, training, compensating, and growth phases of a green workforce, which respects and supports environmentally sustainable principles, activities, and initiatives [73]. Green HRM activities improve the employees' understanding of the environment and allow the implementation of this understanding to accomplish corporate objectives, which contribute to environmentally sustainable workplace behaviour [74]. Meanwhile, PEBs are workplace behaviours that are intended for positive actions towards the environment, such as conserving water and saving energy [75].

Pinzone, Guerci, Lettieri, and Huisingh [76]) and Pham, Tučková, and Jabbour [77] offered empirical evidence that green HRM had a significant impact on PEB. Meanwhile, Saeed, Afsar, Hafeez, Khan, Tahir, and Afridi [38] indicated that green HRM activities promoted green/environmental understanding among employees and improved their actions to develop pro-environmental habits in their personal and professional lives. Pinzone et al. [76] argued that besides the daily social interactions between the employees, the OCBEs conducted by them could be readily witnessed on a day-to-day basis, with

the consequent development of common environmental quality principles and stronger mutual support of OCBEs. The study by Ansari et al. [16] indicates that green HRM practices include setting green responsibilities, targets, and goals, and planning corporate environmental management activities and initiatives, and encourages employees to engage in green behaviours.

The adoption of green HRM strategies in the organisation would improve the employees' environmental consciousness and their ability to conduct environmental behaviour, allow the direct employees to develop shared sustainability philosophy and principles, strengthen business unity, and successfully foster a "climate factory" in the enterprise [78]. A formalised and publicly articulated collection of green HRM practices and policies proved the dedication of the organisation in implementing green practices among the employees. It would also lead to the employees' behaviours to align with the green policies of the organisation [79,80]. Furthermore, green HRM also enhances environment-related performance and generates opportunities for the employees to participate and get involved in organisational green programs [16]. Recently, Elshaer et al. [75] elaborated that employees are expected to exhibit pro-environmental behaviour, whether formal tasks related to the job or voluntary actions, e.g., taking green initiatives when they receive appropriate green HRM. Based on the arguments, the following hypothesis was developed:

**Hypothesis 5 (H5).** *Green HRM has a positive effect on PEB.*

On the basis of previous explanations, the research framework of this study is seen in Figure 1 below:

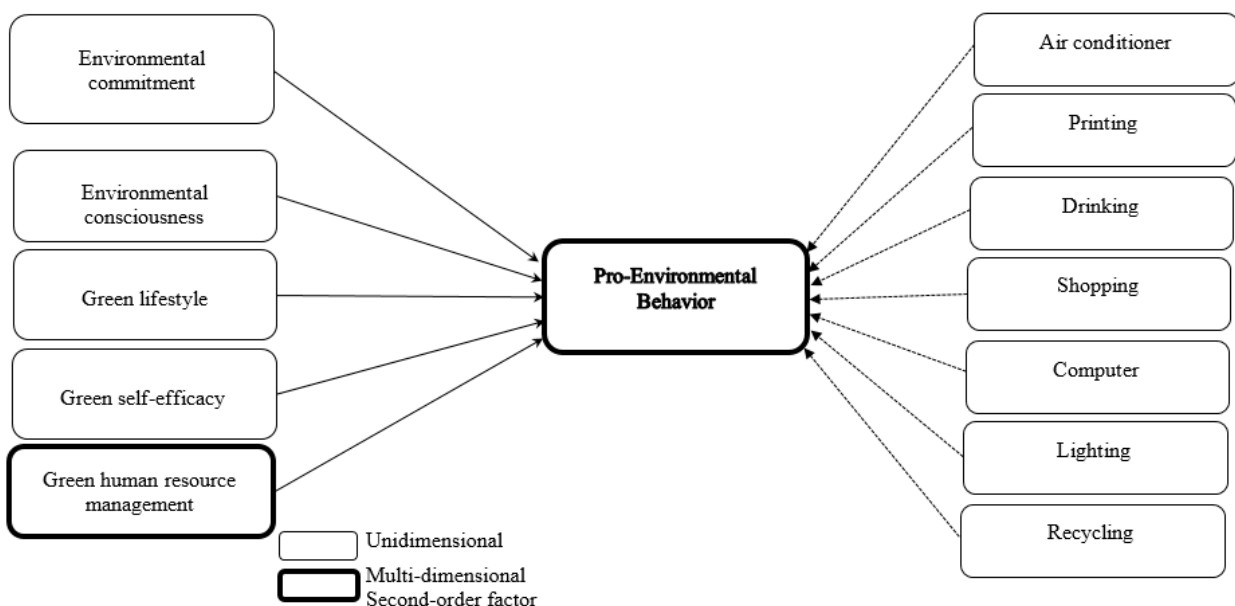

**Figure 1.** Research framework.

## 3. Materials and Methods

### 3.1. Population and Sample

This study was conducted in an organisation in Terengganu, Malaysia. A convenient sampling technique was implemented to collect data from 150 public employees from different levels and departments. A total of 150 questionnaires were distributed, which presented the measures to be rated by the employees. Out of 150 questionnaires distributed overall, 84 usable responses were received, which were represented by a 56% response rate. Due to the difficulty in surveying these respondents based on their roles in the organization, we believe 84 respondents is deemed sufficient. Further, based on the 10-times rule method by Hair et al. [81], which is based on the rule that the sample size should be greater than

10 times the maximum number of inner or outer model links pointing at any latent variable in the model [82]. There are 5 variables pointing at the endogenous construct in the framework, $5 \times 10 = 50$. According to the minimum R2 method, Hair et al. [83] posits that a study with 5 independent variables can have a minimum sample size of 81, with an effect size of 0.1, and power of 80%. Hence, 84 respondents are deemed sufficient for this study. The profile of the respondents is presented in Table 1 below:

**Table 1.** Demography Profile of Respondents.

| Demographic Variables | Frequency | Percentage (%) |
|:---:|:---:|:---:|
| Gender | | |
| Male | 25 | 29.8 |
| Female | 59 | 70.2 |
| Age | | |
| 24–28 years | 12 | 14.3 |
| 29–33 years | 19 | 22.6 |
| 34–38 years | 18 | 21.4 |
| 39–43 years | 11 | 13.1 |
| 44–48 years | 7 | 8.3 |
| 49–53 years | 14 | 16.7 |
| More than 53 years | 3 | 3.6 |
| Marital Status | | |
| Single | 15 | 17.9 |
| Married | 68 | 81 |
| Divorced | 1 | 1.2 |
| Race | | |
| Malay | 81 | 96.4 |
| Chinese | 1 | 1.2 |
| Others | 2 | 2.4 |
| Educational level | | |
| Malaysia Certificate of Education | 6 | 7.1 |
| Diploma | 54 | 64.3 |
| Bachelor's degree | 18 | 21.4 |
| Master | 5 | 6 |
| PhD | 1 | 1.2 |
| Monthly income | | |
| Less than RM2000 | 36 | 42.9 |
| RM2000–RM4000 | 44 | 52.4 |
| RM4001–RM6000 | 3 | 3.6 |
| RM60,001–RM8000 | 1 | 1.2 |
| Status | | |
| Permanent | 28 | 33.3 |
| Contract | 12 | 14.3 |
| Part-Time | 1 | 1.2 |
| Temporary | 43 | 51.2 |
| Tenure | | |
| Less than one year | 16 | 19 |
| 1–5 years | 5 | 6 |
| 6–10 years | 25 | 29.8 |
| 11–15 years | 16 | 19 |
| 16–20 years | 2 | 2.4 |
| More than 20 years | 20 | 23.8 |

*3.2. Instruments*

3.2.1. Pro-Environmental Behaviour

The PEB construct was adapted from Blok, Wesselink, Studynka, and Kemp [5]. It consisted of several sample items, namely (1) "I make sure that air-conditioning is off

or increase the temperature outside working hours" (air-conditioning), (2) "I try to get as much as possible on one sheet (e.g., using narrow margins or printing two pages on one A4 sheet)" (printing), (3) "I sustainably wash the mug (e.g., cold water, no use of washing-up liquids)" (drinking), (4) "When I purchase goods or services, I pay attention to sustainability" (sustainable shopping), (5) "I switch off my computer/notebook when I leave my office for a considerable period" (computer use), (6) "I switch on the lights when I come to the office in the morning" (light use), and (7) "to what extent do you recycle plastic bottles?" (recycling). The measure consisted of 26 items based on a six-point scale from 0 ("not available"), 1 ("never"), to 5 (always).

### 3.2.2. Environmental Commitment

The environmental commitment scale developed by Raineri and Paillé [84] was used in this study. It consisted of several sample items, namely (1) "I care about the environmental concern of my organisation" and (2) "I would feel guilty about not supporting the environmental efforts of my organisation". The measure comprised eight items based on a five-point scale ranging from one ("strongly disagree") to five ("strongly agree").

### 3.2.3. Environmental Consciousness

The environmental consciousness scale developed by Ahmed, Montagno, and Firenze [85], Naffziger, Ahmed, and Montagno [86], Schlegelmilch, Bohlen, and Diamantopoulos [87], and Chang and Chen [88] were applied in this study. This scale consisted of four items, with several items, namely (1) "the employees in the organisation understand the organisation environmental policies and environmental regulations" and (2) "the managers in the organisation are in charge of environmental policies". Each item was rated on a five-point scale ranging from one ("strongly disagree") to five ("strongly agree").

### 3.2.4. Green Lifestyle

This variable was measured a seven-item scale developed by Pickett-Baker and Ozaki (2008) and Sony and Ferguson (2017). In this case, a five-point Likert-type scale ranging from one ("never") to five ("always") was used.

### 3.2.5. Green Self-Efficacy

Green self-efficacy was measured with six items adapted from the study by Chen, Lin, and Weng [89], which were rated on a five-point scale ranging from one ("strongly disagree") to five ("strongly agree").

### 3.2.6. Green Human Resource Management

Six dimensions were used to measure green HRM practices, namely green analysis, description of job position (three items), green recruitment (two items), green selection (two items), green training (three items), green performance assessment (three items), and green rewards (two items). These measurement items, which were adapted from Jabbour [90] and Yong and Mohd-Yusoff [91], were rephrased to represent the individual unit of analysis. To answer each item, a five-point Likert-type scale ranging from one ("not at all") to five ("to a very great extent") was applied.

### 3.3. Data Analysis

This study applies a second-generation structural equation modelling (SEM) approach, namely SEM-Partial Least Square [92]. In this case, a two-step approach was implemented by assessing the measurement model (validity and reliability of the constructs), followed by the assessment of the study hypotheses using the structural model. The data analysed by assisting the statistical software, i.e., SmartPLS 3.3.2.

## 4. Results

Before embarking on hypothesis testing, this study reports the result of the measurement model, which consists of the loading value, construct validity and reliability, and discriminant validity.

### 4.1. Measurement Model

Measurement model quality could be assessed using the convergent and discriminant validity, including the loadings, average variance extracted (AVE), and composite reliability (CR) as per Hair, Howard, and Nitzl's suggestion [93]. It was proposed that the cut-off values for loadings should amount to $\geq 0.7$, AVE $\geq 0.5$, and CR $\geq 0.7$ [94]. As indicated in Table 2, all the loadings were $\geq 0.7$, AVE $\geq 0.5$, and CR $\geq 0.7$, which indicated sufficient convergent validity and reliability in the measurement. In the case of green HRM, which was a second-order reflective measurement (Type I), the first order validity and reliability was evaluated before the assessment of the second-order measurement model validity and reliability, although the first-order loadings were not shown to reduce table length.

**Table 2.** Results of Construct Validity and Reliability (Reflective Model).

| Construct | Item | Loadings | CR | AVE |
|---|---|---|---|---|
| Environmental Commitment | EC1 | 0.734 | 0.962 | 0.759 |
| | EC2 | 0.893 | | |
| | EC3 | 0.864 | | |
| | EC4 | 0.905 | | |
| | EC5 | 0.867 | | |
| | EC6 | 0.879 | | |
| | EC7 | 0.896 | | |
| | EC8 | 0.92 | | |
| Environmental Consciousness | ECN1 | 0.899 | 0.921 | 0.745 |
| | ECN2 | 0.895 | | |
| | ECN3 | 0.884 | | |
| | ECN4 | 0.769 | | |
| Green Lifestyle | GL1 | 0.798 | 0.899 | 0.563 |
| | GL2 | 0.733 | | |
| | GL3 | 0.797 | | |
| | GL4 | 0.726 | | |
| | GL5 | 0.565 | | |
| | GL6 | 0.882 | | |
| | GL7 | 0.712 | | |
| Green Self-Efficacy | GSE1 | 0.737 | 0.927 | 0.68 |
| | GSE2 | 0.85 | | |
| | GSE3 | 0.861 | | |
| | GSE4 | 0.84 | | |
| | GSE5 | 0.767 | | |
| | GSE6 | 0.882 | | |
| Green HRM | Job description | 0.727 | 0.914 | 0.64 |
| | Hiring | 0.719 | | |
| | Selection | 0.791 | | |
| | Training | 0.863 | | |
| | Evaluation | 0.891 | | |
| | Compensation | 0.793 | | |

Provided that a formative measure was performed on PEB, the weights, t-values, *p*-values, and VIF were applied as the standard methods of assessing the quality of the formative measurement items. Besides the significant weights shown in Table 3, no multicollinearity issue was present as the VIFs values were lower than 5 [94]. Therefore, the positive performance was observed from the formative measurement.

**Table 3.** Results of quality assessment of the formative measurement.

| Construct | Dimensions | Weights | t-Value | *p*-Values | VIF |
|---|---|---|---|---|---|
| | Air conditioning | 0.132 | 2.296 | 0.022 | 1.259 |
| | Computer | 0.195 | 6.243 | 0.001 | 1.628 |
| | Drink | 0.101 | 3.616 | 0.001 | 1.487 |
| PEB | Lights | 0.157 | 3.363 | 0.001 | 1.613 |
| | Printing | 0.377 | 5.574 | 0.001 | 1.804 |
| | Purchase online | 0.251 | 7.127 | 0.001 | 2.596 |
| | Recycle | 0.359 | 6.24 | 0.001 | 1.664 |

Discriminant validity was assessed based on the suggestions by Franke and Sarstedt [95] when observing the HTMT ratio. Distinct measures would be developed in the HTMT ratios were lower than 0.85 or 0.90. However, the measures would not be distinct if they were higher compared to the cut-off values. Provided that the HTMT ratios were lower than 0.85 (see Table 4), it was indicated that the respondents clearly understood that six distinct constructs were present in this study.

**Table 4.** Results of Discriminant Validity (HTMT Ratio).

| Construct | 1 | 2 | 3 | 4 | 5 | 6 |
|---|---|---|---|---|---|---|
| 1. Environmental Commitment | | | | | | |
| 2. Environmental Consciousness | 0.391 | | | | | |
| 3. Green Lifestyle | 0.296 | 0.401 | | | | |
| 4. Green Self-Efficacy | 0.211 | 0.5 | 0.201 | | | |
| 5. Green HRM | 0.33 | 0.62 | 0.411 | 0.557 | | |
| 6. PEB | 0.291 | 0.515 | 0.3 | 0.712 | 0.508 | |

*4.2. Structural Model*

To examine the hypotheses developed in this study, a bootstrap with 5000 resampling was operated by Hair et al. [93] and Ramayah et al. [94] to generate the beta values, standard errors, t-values, *p*-values, and confidence intervals. The R2 amounted to 0.557 (Q2 = 0.509), which indicated that the predictors could indicate 55.7% of the variance for PEB. Table 5 shows that Green Lifestyle (β = 0.594, *p* < 0.01) was positively related to PEB. This means that by assuming an increase in Green Lifestyle by 1%, it would give an effect of increasing PEB by as much as 59.4%. The other four predictors were not significant. Therefore, H3 was the only hypothesis being supported, while H1, H2, H4, and H5 were not supported.

**Table 5.** Results of hypothesis testing.

| Hypothesis | Relationship | Std Beta | Std Error | t-Values | *p*-Values | BCI LL | BCI UL | $f^2$ |
|---|---|---|---|---|---|---|---|---|
| H1 | EC → PEB | 0.077 | 0.094 | 0.822 | 0.206 | −0.08 | 0.227 | 0.011 |
| H2 | ECN → PEB | 0.13 | 0.11 | 1.185 | 0.118 | −0.051 | 0.31 | 0.02 |
| H3 | GL → PEB | 0.594 | 0.106 | 5.593 | 0.000 | 0.431 | 0.773 | 0.514 |
| H4 | GSE → PEB | 0.033 | 0.128 | 0.256 | 0.399 | −0.164 | 0.246 | 0.001 |
| H5 | Green HRM → PEB | 0.092 | 0.086 | 1.076 | 0.141 | −0.056 | 0.229 | 0.015 |

## 5. Discussion

This study discusses the relationship between environmental commitment, environmental consciousness, green lifestyle, green self-efficacy, green HRM, and PEB. This section focuses on the main findings of the study, with an emphasis on the implications, limitations of the research, and future research directions. The objective of this study was to assess the elements influencing the PEB of employees in the workplace. This work adds original evidence to the body of knowledge on the adoption of PEB considering the norm-activation model (NAM) [29]. First, the findings revealed that only the green lifestyle had a significant positive relationship with PEB. This result is in accordance with Binder and Blakenberg [58] and Fraj and Martinez [62], who demonstrated the influence of green lifestyle on PEB.

Previous studies found that sustainable development required the systemic participation of the individual in the form of PEB. Hence, the participation in a pro-environmental lifestyle would be one of the many important sectors of lifestyle for an individual [55]. This may imply that there could also be a spill over of green practices from an employee's lifestyle into his work life [61]. In this case, a green lifestyle was involved in terms of consumption, the management of waste, energy-saving, and water conservation [58].

Second, the current study focused on green self-efficacy among employees in a public organisation in Malaysia and extended the empirical literature by examining the impacts of green self-efficacy on PEB. The finding revealed an insignificant effect of green self-efficacy on PEB. This finding was inconsistent with the past research studies (e.g., Choong et al. [72]; Huang [67]; Lauren et al. [64]; Tabernero and Hernandez [70]). The research of Lauren et al. [64] indicated that higher motivation and commitment to participate in these two elements as individuals experience higher self-efficacy concerning PEBs. Hence, Singh et al. [2] added that employees require mastery experience to develop and maintain high levels of self-efficacy. Concerning this, Kim, Kim, Choi, and Phetvaroon [96] suggested the exploration of self-efficacy as a potential determinant of eco-friendly behaviour. Generally, individuals with a high level of confidence in their abilities often exhibit eco-friendly behaviour [68]. Furthermore, Pradhan et al. [66] found a positive impact of self-efficacy on PEB, in which self-efficacy in the public and private manufacturing industries in India was measured. This could mean that in this study, employees have not attained enough knowledge to give them that level of confidence to practice PEB. Hence, high self-efficacy, or an increased belief in one's capacity to control events in one's environment, may tend to increase the practice of workplace behaviour [2].

Third, contradictory to previous findings of Afsar and Umrani [40], Ito et al. [48] and Melo et al. [49], this study did not find a positive relationship between environmental commitment and PEB. The study of Melo et al. [49] captured the general pro-environmental attitudes or values of individuals with a measure of environmental self-perception of individuals regarding lifestyles and behaviours. The core principle of the philosophy of environmental responsibility suggests that individuals with a strong degree of interest possess PEBs and pro-environmental aspirations behave on a highly general basis towards the environment. Subsequently, this common regard for the environment directs the expectations of decision making, which contributes to similar environmental behaviour [46]. In addition, Afsar and Umrani [40] elaborated on employees' beliefs that a socially responsible organisation could encourage commitment to the environment and PEB. Hence, the natural environment may be conceived as a psychological commitment target by which an employee expresses his or her sense of responsibility toward sustainability issues [44]. Therefore, environmental commitment will play a key effect in improving both environmental and business performance through promoting the implementation of sustainable practices [97].

Next, although environmental consciousness was claimed to be a significant predictor of PEB, this study found that the influence of consciousness was not significant. This finding contrasted with the findings in some studies (e.g., Cheema et al. [37]; Thormann and Wicker [52]; Zientara and Zamojska [53]) in which environmental consciousness was positively related to PEB. Previous studies indicated that if the degree of exposure to the natural environment is low, an individual may not understand the impacts of environmental destruction on the society and the Earth. This situation would also create employees with poor environmental literacy who are not able to recognise the importance of environmental concerns. Subsequently, despite the employees' participation in CSR programmes, they may not be involved in PEBs. Moreover, Jain et al. [50] found a positive impact of environmental consciousness on PEB, where people with high environmental consciousness would contribute to environmentally friendly behaviour. These results were also in line with the study by Golob and Kronegger [54].

Finally, green HRM practices aim to improve the environmental knowledge of employees to perform PEB in the workplace without hesitation [54]. It was also found that the

influence of green HRM on PEB was not as significant as predicted. This study's finding aligns with previous discoveries [19,74]. Fawehinmi et al. [19] revealed that green HRM does not directly influence PEB. Green HRM has to influence employees' sense of moral obligation toward the environment and impact employees' environmental knowledge to make employees practice PEB [19,74]. This finding was also not in line with the results of Saeed et al. [38], which recorded the resonant activities by the employees when the company incorporated green activities into the HR policy, which were in line with the green policies of the organisation. Several studies suggested that green HRM had a positive influence on PEB (e.g., Elshaer et al. [75]; Zhang et al. [78]; Pinzone et al. [98]; Pinzone et al. [76]; Pham et al. [77]). The study by Elshaer, Sobaih, Aliedan, and Azzaz [75] indicates that the organisation encourages their employees to provide suggestions and initiatives in environmental improvements, and employees are more likely to become environmentally friendly and work in a team to resolve any environmental issues. This is because when firms genuinely invest in green HRM, they make sincere efforts in hiring, educating, and motivating the workforce regarding green initiatives and practices, then their workforce [16]. Therefore, green HRM would promote the execution of green activities by workers and allow employees to play an active role in green actions in the workplace [79].

Rational logic suggests that individuals with high environmental commitment, environmental consciousness, green self-efficacy, and green HRM often engage in PEB [70]. However, this notion was not supported in the current study as environmental commitment, environmental consciousness, green self-efficacy, and green HRM did not influence PEB. To illustrate, the individuals' level of commitment, consciousness, self-efficacy, and HRM to protect the environment might have been weak. Consequently, employees in the organisation assumed that environmental protection was not their concern.

## 6. Conclusions

In conclusion, this study has successfully examined the elements that influence pro-environmental behaviour of employees in the workplace. The findings indicated that a green lifestyle positively influenced PEB. Therefore, the impacts of environmental commitment, environmental consciousness, green self-efficacy, and green human resource management were not significant. With today's status of the world, green practices will be highly appreciated in most industries, especially those in developing countries, such as Malaysia. Some employees are attracted to work for companies with green practices, which may make the implementation of PEB a motivating factor for employees to perform better. In addition, the development of an employee's green lifestyle may aid with their performance back at work. Encouraging employees to apply green practices in their lives will help them become environmentally concerned, contributing to the preservation of the environment, and may improve their individual job performance. To increase environmental commitment, environmental consciousness, green self-efficacy, and green human resource management among employees, organizations should regularly and frequently communicate about the steps taken to protect the environment. In addition, organizations should incorporate environmentally friendly behaviour and commitment to green performance management in green HRM practices. Finally, top management needs to be fully involved in the proper implementation and processes of green HRM practices to encourage employees and make green HRM practices more effective and impactful in the organization.

### 6.1. Theoretical Implications

To attain sustainable development, it is essential to create awareness of this concept in people around the world and foster a more pro-environmental behaviour in the workplace. The study findings provide valuable theoretical insights to the existing literature on employee PEB. Through the lens of the Norm-Activation Model (NAM), this study investigated the connection between environmental commitment, environmental consciousness, green lifestyle, green self-efficacy, green HRM, and employees' PEB. The results revealed

that only green lifestyles had a positive direct effect on PEB. This finding shed light on the need for predictors of PEB to form a norm (personal or social) before influencing PEB. Hence, this extends the importance of normative constructs such as a green lifestyle. This result can be used to extend the literature on the Norm-Activation Model. The results of this study are supported by Han and Hyun [30]; they posited that the Norm-Activation Model plays a major role in the identification of environmentally responsible decision-making and practices. This study contributes to the knowledge of NAM theory by looking at how a green lifestyle can play a significant role in increasing the predicting power of environmental commitment, environmental consciousness, green self-efficacy, and green HRM with employees' PEB.

In addition, this study validates instruments of environmental commitment, environmental consciousness, green lifestyle, green self-efficacy, green HRM and employees' PEB in developing economies, such as Malaysia, as these scales were developed and validated in western countries. Besides, raising awareness about sustainable development and examining the antecedents of PEB through the lens of the NAM will be a contribution to knowledge. Finally, insignificant results require further investigation to add to the discussion and provide new insights in the field of employee PEB. Therefore, future research could further investigate the links between environmental commitment, environmental consciousness, green self-efficacy, green HRM, and employees' PEB in different contexts to establish and confirm synergies and contradictions.

### 6.2. Practical Implications

Several practical implications have been developed in this study, which focused on only one organisation. Nevertheless, this study offered relevant contributions to other organisations, including private organisations (e.g., Cheema et al. [37]; Zientara and Zamojska [53]), the diverse industry sector (e.g., Saeed et al. [38]), marketing companies (e.g., Ito et al. [48]), schools (e.g., Choong et al. [72]), and higher education institutions (e.g., Dono et al. [99]; Fawehinmi et al. [74]). From a practical point of view, since the Norm-Activation Model is applicable in the case of PEB in the workplace, managers should focus on the factors that increase the PEB of employees. Evidence from our research suggests that a green lifestyle contributes to an employee's pro-environmental behaviour. Apart from the individual's personal lifestyle, it is important to note that organizations should establish a green lifestyle in the workplace, such as initiate an office recycling program or encourage volunteering for an environmental group and other environmental activities. Embracing a green lifestyle at work will further strengthen employees' belief in improving the environment.

In addition to establishing a green lifestyle in the workplace, companies should also create environmentally friendly products (such as environmentally friendly cars or energy-saving light bulbs) to help everyone live a green lifestyle and a sustainable life. Further, the top management should engage more in communicating the importance of being committed to the environment to practice PEB. Employees should be empowered to "own" the process of corporate greening through their involvement in green initiative decision making. More importantly, green HRM practices aimed at reducing carbon emissions through employees' dedicated actions (PEB) should be implemented and fully supported by the top management through visible actions. In view of this, the guidance and support of policymakers are essential for the transition to a green lifestyle. The government should promote the concept of a "green lifestyle" through green campaigns and provide incentives for companies to encourage green growth initiatives. It is hoped that with all the support given, the public can embrace a green lifestyle—from home to the work environment with all the support given.

### 6.3. Limitations and Future Research Directions

Despite the strong emphasis placed by the current study on the optimisation of study, several limitations are present. The limitation could be seen from the survey in this study

(e.g., Afsar and Umrani [40]; Cheema et al. [37]; Saeed et al. [38]; Ito et al. [48]), followed by the questionnaire, which only consisted of six variables. The common method in this study also impacted the findings, as all variables were self-rated by the same respondents. Accordingly, future research should focus on replicating the application of the questionnaire used in this study in a large sample size; expanding the questionnaire into different items of PEB. Further, this study used a cross-sectional, quantitative study; future studies may conduct a longitudinal, qualitative study to show a more detailed understanding of the organizational actual green practices and the employees' PEB. Furthermore, considering the insignificant result in this study, future studies should investigate the mediating and moderating variables (e.g., green lifestyle, green passion, green personal moral norms, and transformational leadership) that could enable the influence of environmental commitment, environmental consciousness, green self-efficacy, and green HRM on PEB.

**Author Contributions:** Conceptualization: B.F., Z.M., J.N.F. and M.Y.Y.; methodology: J.N.F., M.Y.Y., J.S., A.u.-H., M.D.J. and T.R.; software: T.R., J.Y.Y., J.S. and M.Y.Y.; validation: M.Y.Y., Z.M., J.S. and T.R.; formal analysis: J.N.F., M.Y.Y., T.R., J.Y.Y., A.u.-H., J.S. and M.D.J.; investigation: B.F., J.N.F., M.Y.Y., J.Y.Y. and Z.M.; resources: J.N.F., M.Y.Y. and Z.M.; data curation: M.Y.Y. and T.R.; writing—original draft preparation: B.F., Z.M., J.N.F., M.Y.Y. and J.Y.Y.; writing—review and editing: M.Y.Y., A.u.-H., J.S., Z.M., J.Y.Y., O.F., B.F., M.D.J. and T.R.; supervision: M.Y.Y.; project administration: Z.M., M.Y.Y., J.S., B.F. and M.D.J.; funding acquisition: Z.M. All authors have read and agreed to the published version of the manuscript.

**Funding:** This research was funded by the Yayasan Ganesha Nusantara, Indonesia through the International Grant in Collaboration with the Universiti Malaysia Terengganu, VOT Number: 53458.

**Institutional Review Board Statement:** The study was conducted according to the guidelines of the Declaration of Helsinki and approved by the Institutional Review Board (or Ethics Committee) of UNIVERSITI MALAYSIA TERENGGANU (UMT) Research Ethics Committee (No. UMT/JKEPM/2020/46 and 18 May 2020).

**Informed Consent Statement:** Informed consent was obtained from all subjects involved in the study.

**Data Availability Statement:** Not applicable.

**Acknowledgments:** The authors would like to thank Yayasan Ganesha Nusantara, Indonesia and Universiti Malaysia Terengganu, Malaysia for supporting this research and publication. We would also like to thank the reviewers for all the constructive comments.

**Conflicts of Interest:** The authors declare no conflict of interest.

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
