# Peer review of "Determinants of Pro-Environmental Behaviour in the Workplace"

_sustainability, doi:10.3390/su14084420_

Round 1

Reviewer 1 Report

The article on pro-environmental behaviours belongs to a growing stream of research on environmental management at company level. The effectiveness of environmental performance largely depends on how employees behave in the workplace (working environment). The aim of the article is '‘to examine the elements influencing the pro-environmental behaviour of employees in  the workplace’. In particular, it was investigated by means of questionnaire surveys according to  norm-activation model (NOP) ‘the relationship between environmental commitment, environmental consciousness, green lifestyle, green self-efficacy, green HRM, and  protecting the environment through human activities (PEB)’. The aim of the research, the way of describing the results obtained, the part devoted to discussion are presented in a clear way.

Areas of weakness:

  • No information concerning the place of investigation – what do we know about it? The only information this is an organisation in Terengganu (line 33). More description is needed. For example - Is it social responsible company?  It should be supplemented with information on the industry, company size, corporate values, number of employees, etc.     
  • Sample size:  84 employees  (line 32); lack of information on the percentage of the whole staff;
  •  As can be surmised, the results of studies indicating that    ‘(…) the impacts of environmental commitment, environmental consciousness, green self-efficacy, and green human resource management were not significant' (line 436-437) may be largely due to the fact that only 33.3% of respondents were permanent workers, and 51.2% were temporary. Nothing is known about the job positions occupied by the investigated employees.  These characteristics have an impact on the general attitude of employees towards their employer and the way HRM affects them. The authors acknowledge the key importance of GHRM by writing that (line 255-257) ‘the adoption of green HRM strategies in the organisation would improve the employees’ environmental consciousness and their ability to conduct environmental behavior …’.  For this reason, more space in the article should be devoted to describing the GHRM in the company/institution under study.

Author Response

Journal Name: Sustainability

Paper Title: Determinants of Pro-Environmental Behaviour in the Workplace

23 March 2022

Dear

Respected Professor

Respected Reviewer

Respected Editor

Journal of Sustainability (Switzerland)

Resubmission of Manuscript

We thank the expert reviewers for their constructive suggestions and comments. Their invaluable suggestions helped us in further improving the outlook of the manuscript: Thank you for providing the opportunity to Our manuscript, “Determinants of Pro-Environmental Behaviour in the Workplace” for the Special Issue “Impact of Industry 4.0 Drivers on the Performance of the Service Sector: Human Resources Management Perspective” in the Sustainability Journal.

We have considered all the suggestions and included it in the revised draft. The separate response sheet to individual comments and suggestions of the reviewers has been attached.

Thank you once again for the opportunity to publish in the reputed ‘Sustainability’ journal’s special issue.

Best Regards,

Professor Dr. Jumadil Saputra (Corresponding author)

Professor Dr. Adnan ul Haque (Section Guest Editor)

Reviewer 2 Report

It's a very clear and coherent text, with internal logic and concern on showing theoretical foundation. In terms of methodology approach, it was impecable, and personally it was very nice to read. But, the main issue/problem in methodology was the sample itself. We are talking about one public institution, were the aswers obtained (number of usable responses, and more importantly the convenient sampling) by the finely built instrument lead to a forced deconstruction of the theoretical construct so much well achieved (theoretical model of analysis ). The sample may not be sufficient, or representative enough to state a non positive correlation between PEB and environmental commitment, environmental consciousness, green self-efficacy, and green HRM. It's the same as stating that PEB it's only achievable by green lifestylers, or that the only way of increasing PEB is by reinforcing green lifestyle in work place.  This statement does not make invalid the propositions made by the authors in «6.2. pratical implications». The suggestion of a qualitative approach makes all sense, or further specifying the minor variables in the research model.

Author Response

(The authors gave the same response as above.)

Reviewer 3 Report

Dear authors,

The paper is very well organized. I recommend the next lines in accordance to improve the quality of the research:

  1. row 36 - needs revision
  2. Introduction section - it must be revised, I recommend to state clearly the focus of the research, thesis, aims, tasks, subject, object, paper structure, etc. The literature review is better to be moved to section 2.
  3. row 218 - revision
  4. Citation - in case the publication is written by more than 2 authors, I recommend to use: Name et al (see row 358). Please revise the citation style in the whole text following the instructions of the journal.
  5. About the investigated company - It is necessary to give detailed explanation about: Why you choose the company?, Which industry the company presents?, Why this company is interesting?; How many employees work at the company? What is the position of the research objects?, etc.
  6. row 321 - revision
  7. The Discussion section must be focused on authors interpretation and results observed by the research. I do not understand what is the purpose of focusing on literature review.
  8. row 422 - revision
  9. row 447 - grammar mistakes

I hope that my recommendations will be helpful to improve the quality of the research.

Author Response

(The authors gave the same response as above.)

Round 2

Reviewer 2 Report

Major improvements have been made on the manuscript.

Still has the word «race» that I recommend to substitute for another.

Author Response

Journal Name: Sustainability

Paper Title: Determinants of Pro-Environmental Behaviour in the Workplace

1 April 2022

Dear

Respected Professor

Respected Reviewer

Respected Editor

Journal of Sustainability (Switzerland)

Resubmission of Manuscript

We thank the expert reviewers for their constructive suggestions and comments. Their invaluable suggestions helped us in further improving the outlook of the manuscript: Thank you for providing the opportunity to Our manuscript, “Determinants of Pro-Environmental Behaviour in the Workplace” for the Special Issue “Impact of Industry 4.0 Drivers on the Performance of the Service Sector: Human Resources Management Perspective” in the Sustainability Journal.

We have considered all the suggestions and included it in the revised draft. The separate response sheet to individual comments and suggestions of the reviewers has been attached.

Thank you once again for the opportunity to publish in the reputed ‘Sustainability’ journal’s special issue.

Best Regards,

Professor Dr. Jumadil Saputra (Corresponding author)

Professor Dr. Adnan ul Haque (Section Guest Editor)
